Disentangling the gut microbiota of Aldabra giant tortoises of different ages and environments

Zakaria Douaa 1 2 3
Sandri Camillo 4
Modesto Monica 5
Spiezio Caterina 4
Scarafile Donatella 5
Cedras Allen 6
Borruso Luigimaria 7
Manghi Paolo 8
Trevisi Paolo 5
Segata Nicola 8
Mattarelli Paola paola.mattarelli@unibo.it 5
Arita Masanori arita@nig.ac.jp 1 2
1 RIKEN Center for Sustainable Resource Science , Yokohama , Kanagawa , Japan
2 Department of Informatics, National Institute of Genetics , Mishima , Shizuoka , Japan
3 Graduate Institute for Advanced Studies, The Graduate University for Advanced Studies, Sokendai , Hayama , Kanagawa , Japan
4 Department of Animal Health Care and Management, Parco Natura Viva - Garda Zoological Park , Bussolengo , Veneto , Italy
5 Department of Agriculture and Food Science, University of Bologna , Bologna , Italy
6 Seychelles Parks and Gardens Authority , Mahé , Seychelles
7 Faculty of Agricultural, Environmental and Food Sciences, Free University of Bozen-Bolzano , Bozen-Bolzano , Italy
8 Department of Cellular, Computational and Integrative Biology, University of Trento , Trento , Italy
Thomas Jonathan
Electronic publication date: 2025 Jun 10
Publication date: 2025
Volume: 13
Electronic Location ID: e19566
Received 2025 Feb 23; Accepted 2025 May 13
Copyright: ©2025 Zakaria et al.
Copyright year: 2025
Copyright holder: Zakaria et al.
License: This is an open access article distributed under the terms of the Creative Commons Attribution License, which permits unrestricted use, distribution, reproduction and adaptation in any medium and for any purpose provided that it is properly attributed. For attribution, the original author(s), title, publication source (PeerJ) and either DOI or URL of the article must be cited.
License URL: https://creativecommons.org/licenses/by/4.0/

Keywords: 16S rRNA, Longevity, Aldabra giant tortoise, Metagenomics, Gut microbiota

Funding: KAKENHI 23H02470, MEXT Japan SOKENDAI Student Dispatch Program 2022-2023 Green Teen Team Foundation Gruppo Contec Ingegneria This work is supported by the authors’ institutions, KAKENHI 23H02470, MEXT Japan (Douaa Zakaria, Masanori Arita), SOKENDAI Student Dispatch Program 2022-2023 (Douaa Zakaria), and Green Teen Team Foundation (Camillo Sandri, Caterina Spiezio). The mission on Curieuse Island was supported by Gruppo Contec Ingegneria. There was no additional external funding received for this study. The funders had no role in study design, data collection and analysis, decision to publish, or preparation of the manuscript.

==============================
Background

The gut microbiota plays a pivotal role in regulating the physiological functions of its host, including immunity, metabolism, and digestion. The impact of environment and age on microbiota can be assessed by observing long-lived animals across different age groups and environments. The Aldabra giant tortoise (Aldabrachelys gigantea) is an ideal species for this study due to its exceptionally long lifespan of over 100 years.

Methods

Using 16S rRNA gene amplicon analysis, we analyzed 52 fecal samples from giant tortoises in Seychelles (Curieuse and Mahé islands) and in a zoological park in Italy, from very young individuals to those of >100 years old. We performed Alpha and Beta diversity analysis, relative abundance analysis, and complex upset plot analysis, comparing the results of tortoises from different environments and age groups.

Results

The diversity and overall composition of the gut microbiota of tortoises were impacted mainly by geolocation rather than their age. The greater diversity of microbiota in wild tortoises was attributed to their food variance such as wild leaves and branches, compared to captive or domesticated conditions. Beta diversity analysis also revealed the contribution of both environment and age to the variation between samples, with environments accounting for a larger proportion of this contribution. Certain bacterial families, such as Spirochaetota and Fibrobacterota, were more prevalent in environments with higher fiber intake, reflecting dietary differences. Additionally, a range of host-independent environmental bacteria was found to be specific to individuals in Curieuse and not in other geolocations. On the other hand, there were no bacterial taxa specific to centenarians, whose microbial complexity was reduced compared to adult or elderly tortoises.

Conclusions

Our records showed that environment is the primary influence in the overall composition and diversity of the gut microbiota of Aldabra giant tortoises. As giant tortoises are amongst the longest-lived vertebrate animals, these findings can be utilized to monitor their health according to their ages, and enhance their conservation efforts.

Introduction

The Aldabra giant tortoise, Aldabrachelys gigantea (Schweigger, 1812), is endemic to Seychelles and native to the Aldabra Atoll (Quesada et al., 2019). The weight can attain a maximum of 250 kg, and its lifespan can exceed 150 years. This species is classified as ‘Vulnerable’ on the Red List of the International Union for Conservation of Nature, indicating a significant risk of extinction in its natural habitat (Baillie, Groombridge & International Union for Conservation of Nature and Natural Resources, 1996). The Aldabra Atoll, recognized as a UNESCO World Heritage Site in the Seychelles, features tortoises that are integral to the preservation of its robust island ecosystem. The relocation of individuals has enabled Aldabra giant tortoises to function as ecosystem engineers, facilitating the restoration of degraded island habitats. Historically, giant tortoises were abundant across the islands of the western Indian Ocean. Subsequent to human settlement, there was a significant decline in their populations, likely attributable to hunting practices and the predation of hatchlings by non-native species such as dogs, rats, and cats. For the sake of conservation, between 1850 and 1990, the remaining individuals on Aldabra Atoll were reintroduced to the Seychelles islands, which historically hosted wild giant tortoises (Gerlach et al., 2013). The analysis of gut microbiota for this specific species presents a promising avenue as a non-invasive method for the assessment of health and wild populations (Sandri et al., 2020).

The gut microbiota, comprising the diverse community of microorganisms inhabiting the digestive tract, plays a pivotal role in host health, influencing immunity, metabolism, and digestion (Diaz & Reese, 2021). Research has established that both age and environmental factors can influence gut microbiota, with numerous observational studies conducted across humans, mammals, and avian species (Hird, 2017). In reptiles, it is evident that environmental factors, including diet and habitat, significantly affect their gut microbiota. A number of studies indicate that wild individuals display greater microbial diversity compared to those in captivity (Tan et al., 2019; Gao et al., 2019). This underscores the importance of examining microbiota in the original, wild environment compared to more artificial environments, including captivity, for assessing animal health and welfare (Odamaki et al., 2016).

Few studies have been conducted on the gut microbiota of reptiles. In the reanalysis of published 16S rRNA sequence data for the gut microbiota of 91 reptile species (Hoffbeck et al., 2023), significant differences were observed in Alpha and Beta diversity. Aside from methodological considerations like sequencing platforms, the predominant factor affecting the microbiota was identified as the host diet, followed by host order, conservation status, environment, and captivity status (Hoffbeck et al., 2023).

Among the reptile species studied, the omnivore Testudines with diets similar to Aldabra giant tortoises were Beale’s eyed turtle (Sacalia bealei) (Fong, Sung & Ding, 2020), Chinese stripe-necked turtle (Mauremys sinensis) (Hong et al., 2021; Khan et al., 2021), red-eared slider turtle (Trachemys scripta elegans) (Peng et al., 2020), and gopher tortoise (Gopherus polyphemus) (Yuan et al., 2015). The research on Beale’s eyed turtle showed significant differences in microbial diversity and composition between captive and wild individuals but had a very small sample size (n = 8) (Fong, Sung & Ding, 2020). Meanwhile, the studies on Chinese stripe-necked turtle (Hong et al., 2021; Khan et al., 2021) and red-eared slider turtle (Peng et al., 2020) focused on the early life stage without considering different environments or age groups. Likewise, the research on gopher tortoise focused on the effect of their kinship (Yuan et al., 2015).

None of the studies had a sample size larger than 50 subjects or considered the influence of age stages on gut microbiota. In our previous research, we reported on the gut microbiota of Aldabra giant tortoises using 17 subjects and focused mostly on the effect of the environment. The study revealed that diet and geographical location significantly influence microbial community composition. However, the small sample size and limited representation across different age stages and environments hindered the ability to draw definitive conclusions regarding the influence of environment and age (Sandri et al., 2020). This limitation highlighted the necessity for a more comprehensive approach, including sampling individuals at various developmental stages, to better understand the relationship between age, gut microbial composition, and longevity.

Indeed, age is an important factor influencing gut microbiota, as observed in mammalian studies (De la Cuesta-Zuluaga et al., 2019; Salazar et al., 2023). In this regard, insights from human metagenomic research provide a valuable framework, as studies have demonstrated distinct microbial profiles associated with different life stages, such as infancy, adulthood, and extreme old age, including centenarians (Biagi et al., 2016).

These studies suggest that microbial communities evolve in response to host physiology and life stage, offering a potential avenue to investigate age-related health and longevity in tortoises. However, Aldabra tortoises possess unique biological and life-history traits that require specialized approaches beyond those typically employed in human studies. For example, Aldabra tortoises exhibit environmental sex determination, where sex is influenced by incubation conditions rather than genetic factors, and they reach sexual maturity relatively late, around the age of 25 years (Swingland & Klemens, 1989). Remarkably, they retain reproductive capacity throughout their exceptionally long lifespans, which can exceed 100 years. This prolonged reproductive potential, combined with their extended development and unique ecological interactions, suggests that their gut microbiota may adapt differently compared to humans or other animals. Consequently, while parallels can be drawn with human studies in terms of sampling across life stages, methodological adjustments are essential to account for the distinct biology and ecological niche of Aldabra tortoises. This tailored approach is critical to uncovering the potential links between the gut microbiota, aging, and the extraordinary longevity of this species.

Addressing the limitations of earlier studies and the need to adapt methodologies to the unique biology of Aldabra giant tortoises, this study aimed to investigate the gut microbiota across different age groups in different environments, including captivity, focusing on microbial composition and diversity that may influence their exceptional longevity. Using next-generation sequencing (NGS) of the 16S rRNA gene, this research sought to identify age-related shifts and environmental variations in the gut microbiota as well as to identify key microbial taxa that may be associated with prolonged health and lifespan in this species. This focused methodology provides a more precise comprehension of the alterations in gut microbiota throughout the various life stages and environmental contexts of Aldabra tortoises. By highlighting shifts in microbial diversity and composition, the findings contribute to bridging the gap between human and wildlife research, advancing our understanding of microbial dynamics in long-lived species and providing a basis for future functional analyses.

Materials & Methods

Fecal sample collection

In this study, 54 samples were obtained from young to centenarian tortoises in four locations in Seychelles and one in Italy (Fig. 1A; Table S1). The samples in Seychelles were collected in February 2020 and in May 2022. In Seychelles, the temperature remains constant throughout the year (27–30 °C). There are two seasons: a wet season from November to March–April and a dry season from April–May until October. In 2022 there was a longer wet season and the flora was not so different between May and the wet season; so we expect no diet variability on Curieuse Island. The diet of tortoises in the controlled environment at Botanical Garden in Mahé is considered stable all the year. In Italy, the sampling was performed in June at the beginning of summer in 2021.

Figure 1 (A) Sampling locations on the world map (from OpenStreetMap; https://www.openstreetmap.org/#map=10/-4.4820/55.5833) and (B) giant tortoise on turf (from Wikipedia; https://en.wikipedia.org/wiki/Aldabra_giant_tortoise#/media/File:Giant_Tortoise.JPG).

Two samples were discarded from the analysis; one was a duplicate obtained from the same tortoise at a different time (1236694F1225703 in Table S1), and the other turned out to have too few reads (1236709F1225719). After this filtering 52 samples from 52 tortoises were retained: 19 from natural habitat on Curieuse, 11 from the nursery on Curieuse (4°16′56.20″S 55°43′59.7″E), 14 from the Botanical Garden in Victoria on Mahé (−4°37′51.60″S 55°27′4.32″E), 3 from a private garden on Mahé, and 5 from Parco Natura Viva—Garda Zoological Park in Verona, Italy (45°28′58.3″N 10°47′42.4″E). Details of these environments including diet are further described below.

As for age grouping, we categorized individuals into four age groups, considering the sexual maturity age of 25 years old (Swingland & Klemens, 1989). There were 17 Juveniles (2–25 years old), 14 Adults (25–70 years old), 15 Elderly (70–99 years old), and six Centenarians (100+). Centenarians (two from Curieuse, one from Botanical Garden, and three from Parco Natura Viva) were separated from the elderly to highlight the environmental differences in aged tortoises.

Accurate measurement of tortoise age in this study was challenging (except for those in the nursery) as estimations were based mainly on body size and shell shape. There are no detailed mortality or disease statistics for tortoises, unlike for humans and other model organisms.

The age and the origin of the juvenile tortoises at Parco Natura Viva were well known and reported in ZIMS species360 international database (also reported in Table S1). The age of the centenarians of Parco Natura Viva was estimated from the very old documents of the park. The age of the private tortoises was also known, and it was told to the authors CS and CS by the owners. Regarding the age of the tortoises on Curieuse, many of them were translocated from Aldabra Atoll in late 1970s when they were adults. The translocated tortoises were tagged with a metal disc attached to the 4th dorsal (D4) carapace, and the disc was renewed in 1997 and in 2013 when necessary (Mason-Parker et al., 2018). The age class of these translocated tortoises were classified as “Elderly” in this work. Many juveniles were presumably stolen, however, and often not found during census (Gerlach et al., 2013). The remaining tortoises were identified with a subjective estimate as “Juveniles” or “Adult.” The authors CS and CS adopted the same subjective estimation to classify the tortoises of the Botanical Garden to the different age classes through comparison with the ones of known age.

Sampling was conducted following the Nagoya Protocol, as previously stated in Sandri et al. (2020). Fresh feces were obtained from each individual tortoise identified by tags or specific carapace characteristics. Disposable sterile gloves were used during sample collection to prevent contamination. The fecal material was extracted from the center of large, fresh, and undamaged pieces to avoid soil contamination. Samples were obtained in the early morning, in the late morning, and in the early afternoon following the activity patterns of the tortoises. The small plastic shovel-like tool attached to the screw-cap tubes was used to scoop the fecal material. Approximately one g of each sample was placed in screw-cap tubes (DNA/RNA Shield Fecal Collection Tube, Zymo Research Europe, Germany). All collected samples were kept in a portable cooler with ice packs or stored in a refrigerator until they reached the laboratory.

Environment and diet of giant tortoises

The age and sampling locations of 52 individuals (when duplication was removed) are provided in Table S1. All individuals were without medication. The ages of adult and older tortoises were estimated in 10 s from accretion lines on their carapace.

Wild environment and nursery on Curieuse

The tortoises in the wild can access the native island vegetation. They are selective grazers, feeding on grass (turf) (Fig. 1B), leaves, woody stems, endemic fruit, especially tropical almonds (Terminalia catalpa), and flowers of the season. In particular, “tortoise turf” is a complex of grasses, herbs, and sedges that the grazing pressure of the tortoises has dwarfed (Merton, Bourn & Hnatiuk, 1976). They can graze freely till near the beach or in the forest, where they can find lignocellulosic materials.

The nursery is dedicated to caring for young tortoises, those up to 6 years old. The facility is overseen by a dedicated team tasked with safeguarding the young tortoises against predators, poaching, and human interference. The diet of these tortoises is meticulously crafted by the staff of the Seychelles National Parks Authority, who gather fresh young leaves and green vegetation from the island. Additionally, their diet is supplemented with commercial fruits every week. Their ages are recorded accurately and they undergo a reintroduction program to acclimatize to wild conditions when they come to age.

Botanical garden and private garden on Mahé

The National Botanical Garden on Mahé provides a spacious habitat for Aldabra giant tortoises, spanning 15 acres and featuring different elevations. This enclosure incorporates various elements such as rocks, a sandy area, water features, and muddy pools to cater to the tortoises’ diverse needs. The primary diet for the tortoises consists of native branches and leaves endemic to the Seychelles, with no grass included. Additionally, a selection of fruits like star fruit, golden apple, jackfruit, mango, and papaya are added. Furthermore, the staff regularly prepares banana leaves (with stem) for visitors to feed the tortoises directly, to add an interactive element to the visitors’ experience. The house of Perley Constance on Mahé, a tour guide and conservationist in Seychelles, keeps a few tortoises in the garden. The diet of these tortoises is also carefully crafted, but the details are less known.

Parco Natura Viva in Verona

The housing at Parco Natura Viva is an enclosure featuring both indoor and outdoor sections. These sections are divided into two zones: one for senior tortoises of two males and a female, all aged over 100 years, and the other for younger ones, approximately 15 years old. The tortoises can enjoy uninterrupted access to ultraviolet and heat lamps, a dedicated pool area, and sandy terrain within their indoor space. They are accommodated indoors during nighttime, in cold weather (when temperatures drop below 18 °C), and throughout the winter season, which spans about five months. For the remainder of the year, they can roam in the expansive outdoor area, encompassing 1,040 square meters. The feeding schedule is four days per week. Their diet comprises a blend of leafy greens and vegetables, primarily consisting of salad and Italian chicory. Additionally, once a week, they are treated to seasonal fruits such as apples and provided with hay. Essential supplements, such as Calcium, are thoughtfully supplied. During the summer months when the sampling was performed, the tortoises had access to a patch of turf with grass, allowing them to graze at their leisure.

DNA extraction and NGS sequencing

DNA extraction, purification, and quantification were performed as previously described in Borruso et al. (2021) and Sandri et al. (2020). The V3–V4 region of the 16S rRNA gene (∼460 bp) was amplified and sequenced using the Illumina MiSeq platform 2 × 300 bp. Gene amplicons were produced using the primers Pro341F: 5′-CCTACGGGNBGCASCAG-3′and Pro805R: 5′GACTACNVGGGTATCTAATCC-3′(Takahashi et al., 2014). PCR and MiSeq sequencing were conducted as previously described in Borruso et al. (2021). The raw reads obtained are publicly available at the Sequence Read Archive (SRA) under the BioProject accession number: PRJNA1076075.

Data processing

Analysis of the raw sequences were conducted with DADA2 (Callahan et al., 2016), DECIPHER (Wright, 2020), Phangorn (Schliep, 2011), and Biostrings (Pagès et al., 2024) packages in R Statistical Software (v4.3.0). For taxonomy assignment, the GTDB database (release 214.1) was used for accurate taxonomic identification from genomes (GTDB is based only on taxa whose genomes are available). Prior to running the DADA2 pipeline, primer sequences were removed from the raw sequences using the cutadapt tool. The expected error threshold (maxEE) parameters of DADA2 were set at 2 for forward reads and 5 for reverse reads. We discarded any reads shorter than 240 bp or longer than 460 bp based on their expected amplicon size. All other parameters used in DADA2 remained at their default settings.

Statistical analysis

The statistical analysis was carried out in R, using the phyloseq, microbiome, and maaslin2 packages (Shetty & Lahti, 2019; McMurdie & Holmes, 2013; Mallick et al., 2021). For the statistical significance test, the Wilcoxon non-parametric test was used in Alpha diversity analysis. Permutation multivariate analysis of variance (PERMANOVA) test was used for Beta diversity analysis to measure statistical significance (Oksanen et al., 2020). Pairwise Adonis test was used for pairwise comparisons using Benjamini–Hochberg method for p-value correction (Arbizu, 2020). To reduce noise in the relative abundance figures, the taxa whose median relative abundance was less than 1% were classified as“Remainder” for all taxonomy levels. For the family and genus names, only those appearing in more than 50% of samples are listed in the figures. All data are retained in the Supplementary Information. Finally, the ComplexUpset package in R was used for the set-intersection Analysis. All analysis steps are described in more detail in our GitHub repository: https://github.com/douaazak/AldabraGiantTortoises_Microbiome_Analaysis.

Results

Relative abundance of bacterial phyla and families

We first estimated the microbial composition using Amplicon Sequence Variants (ASVs). A total of 1,749,265 high-quality paired sequences were filtered from 3,870,949 reads obtained from the 52 samples across five locations: Curieuse, Curieuse (Nursery), Private Garden, Botanical Garden, and Parco Natura Viva, yielding an average of 33,639 reads per site. The sequences were then grouped into 4,866 ASVs and classified into 30 phyla, 68 classes, 270 families, and 580 genera with GTDB database (Table S2).

A mere 6.9% of all ASVs were able to acquire genus identifiers. When considering only ASVs with an abundance exceeding 1%, the number of remaining ASVs was only 230, and 64 genera were identified. The identifiers for the genus in this context include informal designations, such as Bact-08 and UBA5429, as illustrated in Table S2 and Fig. S1. The limitation to standard scientific nomenclature resulted in a reduction of nearly fifty percent in the number of genera. Given the extensive number of ASVs belonging to unidentified genera, we were unable to discuss compositional differences at the genus level; therefore, we focused our analysis on the phylum and family levels. While the GTDB database offered robust taxonomic classification based on whole genomes, the high ratio of unclassified taxa in our analysis may reflect the absence of ribosomal operons in possibly incomplete genomes in the database.

Figure 2 illustrates the relative abundance plots for the most prevalent phyla and families. Bacillota (previously known as Firmicutes) and Bacteroidota (formerly referred to as Bacteroidetes) were predominant across all groups, consistent with our earlier findings (Sandri et al., 2020). The extent of this dominance was notably greater in the Private Garden and Parco Natura Viva, attributed to their restricted food variety. Besides, fecal samples from Parco Natura Viva exhibited a markedly smaller number of taxa.

Figure 2 Relative abundance of the (A) bacterial phyla and (B) families having >0.1% abundance.

The samples are grouped by locations and are sorted by estimated age from left to right in each geolocational group.

At both the phylum and family levels, samples from the same location exhibited similar composition (Fig. 2). At the family level, Clostridiaceae (17% on average), Lachnospiraceae (14%), Peptostreptococcaceae (1.8%), Ruminococcaceae (1.8%), and Acutalibacteraceae (1.8%) were dominant in all locations, and these families are recognized for their association with plant polysaccharides and fibers. The samples collected from Curieuse and the Botanical Garden encompassed various age groups; however, no statistically significant differences were detected among the age groups within each respective environment.

The differential abundance analysis conducted using Maaslin2 at the phylum level revealed a greater number of phyla and families present in the wild Curieuse compared to the other environments. For example, bacterial families often inhabiting the gut of herbivores such as Ruminococcaceae, Oscillospiraceae, and Treponemaceae were better represented in the wild Curieuse tortoises.

Upon comparison of age groups, it was observed that juvenile tortoises exhibited a reduced abundance of these families, suggesting that wild adult and senior tortoises obtain these bacterial families from their environment (Fig. S2).

Figure 3 Alpha diversity (Shannon index) in (A) different environments and (B) age groups.

The sample colors indicate their groupings. The pairwise Wilcoxon test was applied. Only significant results (p-value < 0.05) are shown.

Diversity analysis

The Shannon index for Alpha diversity was initially calculated to quantitatively evaluate the diversity (Fig. 3A). Upon examining the variations in Alpha diversity across different environments, it was observed that the samples from Curieuse Island demonstrated a markedly higher level of diversity compared to the other environments, with the exception of the Botanical Garden on Mahé. Additional indices for Alpha diversity, specifically Chao1 and Simpson, yielded consistent results (Fig. S3). In terms of age groups, juvenile tortoises (2 to 25 years old) exhibited significantly lower diversity than adult and elderly tortoises, but not centenarians (Fig. 3B). The apparent resemblance between juveniles and centenarians can be attributed to the limited group size of centenarians and the notably low Shannon indices observed in individuals from Parco Natura Viva, applicable to both juveniles and centenarians.

Subsequently, the weighted UniFrac, Bray–Curtis, and unweighted UniFrac distances were computed utilizing the 4,866 ASVs to assess Beta diversity across various age groups and environments in an unsupervised approach. The weighted UniFrac PCoA analysis showed that data points were clustered by geolocation rather than age (Fig. 4). The permutation multivariate analysis of variance (PERMANOVA) test showed that the coefficient of determination or the R-squared score was 46.2% for environmental variation, while 24.5% for age groups. The pairwise PERMANOVA analysis indicated notable differences across all environments. At the level of age groups, the pairwise comparisons revealed significance solely between Juveniles and Adults, as well as between Juveniles and Elderly tortoises (Table S3). Similar clustering and PERMANOVA test results were obtained using Bray–Curtis (38% for environments, and 16.8% for age group) and unweighted UniFrac distances (32.2% for environment, and 16.4% for age group) (Fig. S4).

Figure 4 PCoA plot for the Beta diversity (Weighted UniFrac).

Sample colors indicate age groups and shapes indicate environments. The PERMANOVA test includes R-squared (R2), P-value, and pseudo F-statistics (F).

Shared and unique taxa among different environments and age groups

To visualize the numbers and the ratio of shared bacterial taxa among different groups, we created the Upset plot at the family and genus levels (Fig. 5). This plot visualizes set intersections as the matrix with connected dots, and their compositions as associated bar charts. The percentage of sharing is much higher among the age groups (43% family and 32% genus shared for all ages) rather than the geolocation groups both at the family and genus levels. The wild environment of Curieuse exhibited high percentages of unique taxa both in families and genera, 12 and 13%, respectively, whereas the captive environment of Parco Natura Viva showed almost no unique taxa. For example, environmental bacteria were unique to samples from Curieuse Island such as Actinomycetaceae, Amphibacillaceae, Azospirillaceae, Bryobacteraceae, Dermabacteraceae, Desulfitobacteriaceae, Haladaptataceae, Methyloligellaceae, Nitrospiraceae, Pyrinomonadaceae and Thermococcaceae, and these were not found in other geolocations. In terms of age, the young generation possessed higher rates of unique taxa (15 and 22%) than the other three age groups. The youth-associated bacterial genera whose abundance exceeded 1% in any individual were Rheinheimera, Pseudomonas, Halopseudomonas, Ureibacillus, Pseudogemmobacter, Methanobrevibacter, Taishania, Cytobacillus, and Advenella. These genera are usually found in different environments (soil, water, or both), and meaningful hypotheses are hard to establish from the 16S sequencing results.

Figure 5 The number and percentages for bacterial families in different geolocation (A) and age groups (C), and bacterial genera in different geolocation (B) and age groups (D).

Colors show taxonomic classification based on phyla. Percentages above bars are rounded to the closest integer, and each of them shows the percentage of items in the intersection relative to the total number of items in all intersections. For clearer visualization, only the top 10 phyla are displayed while the rest is named as remainder, and the number of intersections is limited to 10.

Discussion

Observing the similarities and differences in the gut microbiota of animals across different environments and age groups is beneficial for getting feedback on the management of the animals under human care and for the conservation of the species. In addition, Aldabra giant tortoise is the longest-living terrestrial species. Understanding the relationship between age and gut microbiota would be advantageous. The findings of our research underscore the considerable impact of geolocation on gut microbiota, revealing that tortoises inhabiting analogous environments exhibit similar composition and diversity indices, irrespective of their age groups.

In terms of Alpha diversity, tortoises in the wild were more diverse than those housed in controlled environments. Variations in diet, exposure to diverse soil and vegetation types, and interactions with other wildlife present in these natural surroundings may all contribute to the observed diversity. Among the bacterial families unique to individuals from the wild Curieuse Island, Bryobacteraceae and Haladaptataceae are characteristic of salty environments, suggesting grazing in coastal areas. Likewise, Azospirillaceae and Nitrospiraceae are often involved in soil nitrogen fixation, suggesting grazing of grass or turf. Presumably they are not the core microbiota of tortoises, and the contribution to tortoises’ health is unknown, but the higher diversity observed on Curieuse Island suggests the large influence of their habitats on their welfare. Fewer differences among age groups support that the gut microbiota of tortoises does not significantly change with age after maturation. Significant differences were found only between growing juvenile tortoises and grown-ups.

Some families exhibited a clear increase or decrease within a smaller group with clearer age determinations. Inside Curieuse Island and Curieuse Nursery, for example, Ruminococcaceae increased in abundance with age; tortoises older than 80 years old had significant increase in their relative abundance in comparison with younger tortoises. Similarly, Lachnospiraceae decreased for 70 years or older tortoises. Both families are known to degrade plant polysaccharides, but the former was associated with terrestrial herbivorous reptiles and the latter with marine reptiles (Campos et al., 2018). Probably, the increase of Ruminococcaceae is linked with a larger ratio of land grasses, leaves, and branches.

Looking at the Beta diversity, PERMANOVA statistical test showed that both age group and geolocation have significant contributions to the variation between samples (p-values equal 0.001 for both factors). The R2 values in weighted UniFrac plot showed that the geolocation had a larger contribution (R2 = 0.462) to the overall variation than the age group (R2 = 0.245). The pairwise PERMANOVA results showed that the contribution of geolocation to the variability between samples was distributed between all geolocations where all pairwise comparisons were significant except for Curieuse (Nursery) vs. Private Garden (Table S3). The lowest R2 values were obtained between Botanical Garden and Curieuse (R2 = 0.121). This is also evident by looking at the clustering of the tortoises from these environments in Fig. 4. Meanwhile, the contribution of age to the variability is largely explained by the differences in the pairs of Juveniles vs. Adults (R2 = 0.209), and Juveniles vs. Elderly (R2 = 0.236).

Moreover, the overall composition of bacterial phyla and families was consistent with the locational grouping regardless of their age. Notably, Spirochaetota was more prevalent in Curieuse Island and Botanical Garden, potentially attributed to higher fiber intake from ingested plant polysaccharides. This is related to the increased level of Spirochaetes, known for producing high levels of short-chain fatty acids, thereby maximizing metabolic energy extraction (Thingholm et al., 2021). Similarly, Fibrobacterota, which includes major bacterial degraders of lignocellulosic material in herbivore gut (Quesada et al., 2019), was more abundant in Curieuse Island and the Botanical Garden where branches and leaves are fed. On the contrary, the presence of Lachnospiraceae, responsible for producing a large amount of butyrate (Vacca et al., 2020), was less common in Curieuse Island and Botanical Garden.

In Parco Natura Viva, Bacillota_A (formerly Firmicutes) was abundant, and a similar abundance was reported in the captive Beale’s eyed turtle (Sacalia bealei) compared to wild turtles. This was attributed to the limited diet that the turtles and the tortoises received in their captive environment (Fong, Sung & Ding, 2020). At the genus level, we observed that Bacteroides and Clostridium exhibited an abundance of 0.04% and 8.7% respectively, in alignment with the reanalysis of 91 reptiles (Hoffbeck et al., 2023), where these genera were reported as the core microbiota of reptiles.

The complex Upset diagrams reinforced the observations that bacterial taxa are more commonly shared across age groups (43% of families and 32% of genera among four groups), compared to geolocations (16% of families and 13% of genera among four locations). Analyzing the bacterial taxa unique to specific sets revealed that Curieuse wild tortoises exhibited the highest proportions of exclusive taxa among environments (12% of exclusive families and 13% of exclusive genera). This result further validates the Alpha diversity results, which indicate that Curieuse has significantly greater microbial diversity than other environments, except Botanical Garden. In contrast, when age groups were examined, juvenile tortoises displayed the highest percentages of exclusive taxa, comprising 15% of families and 22% of genera. This trend was not reflected in their Alpha diversity metrics. This may be due to the uneven distribution and likely low abundance of these additional bacterial families and genera, which were absent in other age groups.

Our current investigation on Aldabra giant tortoises yielded results that align with our previous study, demonstrating consistency in dominant taxa across phylum, family, and genus levels, alongside the environmental contribution to diversity among samples. Moreover, although Sandri et al. (2020) offered initial descriptive insights into the gut microbiota of Aldabra giant tortoises, their limited sample size may have introduced biases in the environmental comparisons. Specifically, the wild Curieuse environment had a larger sample size than the other environments with significant within-group variations. This likely accounts for the scattering of Curieuse data for Alpha and Beta diversity analyses in our previous study (Sandri et al., 2020).

On the other hand, this study utilizes a larger and more diverse set of samples, encompassing various age groups and environments, allowing for identifying distinct clustering of Curieuse samples and their higher Alpha diversity. Additionally, we identified numerous differentially abundant and unique taxa in the Curieuse tortoises, highlighting the distinction between wild and captive conditions. Although diet was not directly assessed in this study, it is apparently a contributing factor to the differences between environments. Furthermore, this study allows for the investigation of age effects, given the large sample size ranging from 2 to over 100 years old. Juvenile tortoises exhibited lower diversity compared to adults and displayed certain differentially abundant taxa. Interestingly, centenarian tortoises also exhibited lower diversity although the number of samples and the environmental distribution are not enough to conclude the shift.

The limitation of this study is the low taxonomic resolution due to the 16S sequencing technology. Even within the same family, the metabolic capability of bacteria may vary. For well-investigated species such as humans and mice, the metabolic roles of common bacteria are often well-grounded. Still, such related information is scarce for reptiles, not to mention giant tortoises. For this reason, future research direction includes longitudinal studies tracking individual tortoises over time, metagenomic analysis for a more comprehensive view of microbial functions, and investigation of host-microbe interactions in relation to tortoise health and longevity. Controlled dietary experiments and comparative studies with other long-lived reptile species could further elucidate the complex interactions between environment, diet, age, and the gut microbiota in these remarkable long-lived reptiles.

Conclusion

Our study revealed that environmental factors, particularly habitat, influence gut microbial composition more strongly than age. Wild tortoises exhibited significantly higher microbial diversity than those in controlled environments, likely due to exposure to more varied food sources and environmental bacteria. Juvenile tortoises exhibited lower Alpha diversity than grown-ups, but clear comparison between age and habitat was difficult for tortoises in captivity. The dominant bacterial phyla across all samples were Bacillota (formerly Firmicutes) and Bacteroidota (formerly Bacteroidetes), which were consistent with previous findings. Certain bacterial families, such as Spirochaetota and Fibrobacterota, were more prevalent in environments with higher fiber intake, reflecting dietary differences. As giant tortoises are amongst the longest-lived vertebrate animals, these findings can be utilized to monitor their health according to their ages, and enhance their conservation efforts. However, the study is limited by challenges in age determination, especially for older wild tortoises, and small sample sizes for some environments and age groups.

Supplemental Information

Supplemental Information 1 The age and sampling locations of 54 individuals

Supplemental Information 2 4,866 ASVs and classified into 30 phyla, 68 classes, 270 families, and 580 genera

Supplemental Information 3 Pairwise PERMANOVA tests

Supplemental Information 4 Relative abundance of the bacterial genera having >0.1% abundance

The samples are grouped by locations and are sorted by age group from left to right in each geolocational group.

Supplemental Information 5 Differential abundance heatmap (Maaslin2) for bacterial phyla, and families

(A) Curieuse as a reference for environments. (B) Juveniles (2–25) as a reference for age groups. Values are for top 50 taxa with significant associations representing (–log(qval)*sign(coeff)).

Supplemental Information 6 Alpha diversity in different environments using Chao1 (A) and Simpson (C), and in different age groups using Chao1 (B), and Simpson (D)

The pairwise Wilcoxon test was applied.

Supplemental Information 7 PCoA plot for the Beta diversity using Bray Curtis (A), and unweighted UniFrac (B)

Sample colors indicate age groups and shapes indicate environments. The PERMANOVA test includes R-squared (R2), P-value, and pseudo F-statistics (F).

Additional Information and Declarations

Competing Interests

Author Contributions

Animal Ethics

Field Study Permissions

Data Availability

The authors declare there are no competing interests.

Douaa Zakaria performed the experiments, analyzed the data, prepared figures and/or tables, authored or reviewed drafts of the article, and approved the final draft.

Camillo Sandri conceived and designed the experiments, analyzed the data, authored or reviewed drafts of the article, sampling design and arrangements including permissions, and approved the final draft.

Monica Modesto conceived and designed the experiments, performed the experiments, authored or reviewed drafts of the article, and approved the final draft.

Caterina Spiezio conceived and designed the experiments, analyzed the data, authored or reviewed drafts of the article, sampling design and arrangements including permissions, and approved the final draft.

Donatella Scarafile performed the experiments, authored or reviewed drafts of the article, and approved the final draft.

Allen Cedras conceived and designed the experiments, authored or reviewed drafts of the article, sampling design and arrangements including permissions, and approved the final draft.

Luigimaria Borruso conceived and designed the experiments, authored or reviewed drafts of the article, and approved the final draft.

Paolo Manghi analyzed the data, authored or reviewed drafts of the article, and approved the final draft.

Paolo Trevisi performed the experiments, authored or reviewed drafts of the article, and approved the final draft.

Nicola Segata analyzed the data, authored or reviewed drafts of the article, and approved the final draft.

Paola Mattarelli conceived and designed the experiments, performed the experiments, analyzed the data, prepared figures and/or tables, authored or reviewed drafts of the article, and approved the final draft.

Masanori Arita performed the experiments, analyzed the data, prepared figures and/or tables, authored or reviewed drafts of the article, and approved the final draft.

The following information was supplied relating to ethical approvals (i.e., approving body and any reference numbers):

Sampling was conducted following the Nagoya Protocol, by the guidelines set forth by the European Commission’s Guidance document on the scope of application and core obligations: Guidance Document on the Scope of Application and Core Obligations of Regulation (EU) No 511/2014 of the European Parliament and of the Council on the Compliance Measures for Users from the Nagoya Protocol on Access to Genetic Resources and the Fair and E. Ocala, FL: EC Publishing.

The following information was supplied relating to field study approvals (i.e., approving body and any reference numbers):

Seychelles Parks and Gardens Authority; Department of Animal Health Care and Management, Parco Natura Viva

The following information was supplied regarding data availability:

The sequences are available at NCBI-SRA: PRJNA1076075. The analysis steps are available at GitHub: https://github.com/douaazak/AldabraGiantTortoises_Microbiome_Analaysis.

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
