# Peer review of "Disentangling the gut microbiota of Aldabra giant tortoises of different ages and environments"

_PeerJ, doi:10.7717/peerj.19566_

## Round 0.1 · original submission · Major Revisions

While the study is of interest and has merit, both reviewers had some concerns that will require addressing before the manuscript can be accepted for publication.

I note that the reviewers do disagree in one place in particular in the methods, specifically around lines 229-232 and the description of samples that were excluded. Personally I thought this was of value and could be addressed as suggested by reviewer two, rather than removal.

Reviewer 1 ·

Basic reporting

Language needs revision

Experimental design

Major concern is the GTDB database used for classification.
Results need to be presented in a clearer way and could be broken down into sections as suggested in the attachment.

Validity of the findings

Main findings that age is less significant than environment and diet seems to be valid. Statistics are done and shown in the figures but need to be brought to the text. Weighted Unifrac PCoA explained more variance and need to be brought to the main figures.

Annotated reviews are not available for download in order to protect the identity of reviewers who chose to remain anonymous.

Reviewer 2 ·

Basic reporting

This article provides a detailed insight into the gut microbiota of Aldabra giant tortoises across different ages and habitats. It builds upon the authors' previous findings published in Sandri et al. (2020), expanding the sample size and focusing on age-related differences in gut microbiota. The central hypothesis is that gut microbiota changes over the tortoise’s long lifespan.

My main concern is that the reported results do not fully reflect this primary objective (assessing the effect of age). Instead, environmental factors are highlighted as the main driver of microbiota differences. Upon first reading, this discrepancy makes it difficult for the reader to understand why the title emphasizes "different ages," while the Abstract, Results, and Discussion sections initially focus on geolocation. If the primary goal is to assess the effect of age, this should be the first aspect reported in the Abstract, Results, and Discussion. While age is found to significantly influence alpha and beta diversity, this is not clearly emphasized in the Abstract or Discussion. Strengthening this emphasis would improve clarity. After presenting the significant effect of age, you can highlight that environment is even stronger predictor of gut microbial community composition.

The Introduction provides a strong background, and the references cited are relevant. The research aims are clearly stated.

Title (Line 2): I suggest modifying the title to “Disentangling the gut microbiota of Aldabra giant tortoises across different ages and habitats.” The term microbiota (rather than microbiome) is more precise, as 16S sequencing identifies microbial taxa rather than their full genetic potential. Additionally, since the study also assesses habitat differences, this should be reflected in the title.

Lines 96–98: Clarify whether “significant differences in diversity and composition” refer to the microbiota or diet.

Experimental design

The experimental design is well-structured, and the methodologies used are appropriate for this type of research, with minor clarifications needed. The raw data are available in the SRA database, and the analysis code is accessible via GitHub. I recommend adding the GitHub repository link in the Data Availability section for completeness.

Specific Comments:

Line 146: The sample numbers should be explained more clearly. You obtained 54 samples from 53 animals (due to one duplicate) and analyzed 52 samples (excluding one with insufficient sequences). This clarification should be included in the main text, along with the sample IDs of the discarded samples.

Lines 147–150: It would be helpful to include the number of animals sampled at each location directly on the map in Figure 1.

Line 167: Clarify whether the total number of individuals is 54 or 53, given that one sample was duplicated. Is this the individual crossed out in Table S1? If so, this should be explicitly stated.

Lines 151–155: Provide references to support the biological reasoning behind the age groups. How was age estimated? How many individuals were in each age category?

Figure 1: I strongly recommend removing the images of food since diet was not directly tested, and the focus is on age (and environmental) differences. More beneficial would be adding images of tortoises of each age category if available. Additionally, ensure that figure labels (a, b, etc.) follow the order in which they are referenced in the text.

Lines 229–232: Combine this section with the overall clarification of sample numbers, explicitly listing the sample IDs of those excluded.

Line 236: Specify what aspect of the analysis was tested using the Wilcoxon non-parametric test—was it for alpha diversity?

Validity of the findings

The findings are valid, and the visualizations are well-annotated, with analyses appropriately performed. However, some parts of the text and figures require improvement for better clarity and presentation.

Line 289: Cite Figure 4 and Supplementary Figure S3 where PERMANOVA results are discussed. Additionally, consider performing pairwise PERMANOVA to identify specific group differences and reporting the results in a table.

Line 292: The figure reference appears to be incorrect. Supplementary Figure S3 relates to alpha diversity, not UniFrac distances.

Figure 3: Adjust the y-axis scale to start from 3 to make differences between categories more visually apparent in the boxplots.

Figure 4: I recommend replacing the unweighted UniFrac PCoA with the Bray-Curtis PCoA from Figure S4b in the main manuscript. Bray-Curtis explains more variability (37% and 17.8%), and juveniles are distinctly separated from other groups along Axis 2, supporting your conclusions effectively. You could also enhance visualization by using colors for age groups and shapes for environments.

Figure S1: The resolution is too low; the genera names are difficult to read.

Line 316: Since diet differences were not directly tested, avoid strongly concluding that dietary variation is a main driver of microbiota changes. Rephrase this statement elsewhere to reflect that diet may be a potential contributing factor, but it was not assessed in this study.

Line 319-320: If age is the primary factor being tested, the discussion should begin with age-related findings rather than environmental influences.

Line 331-332: This should be stated in Methods section, along with exact methodology how the age was inferred and adding the relevant references of the methodology if it exists.

Line 335-336: More discussion should be provided on the contribution of age within smaller location-based groups.

Line 344: Include discussion of pairwise PERMANOVA results here.

Line 403: The Conclusion should first address how age influences tortoise gut microbiota before discussing environmental effects. Since diet was not directly tested or reported, it should be removed from the Conclusion.

Additional comments

Final Remarks
This study is well-designed and contributes valuable insights into the gut microbiota of Aldabra giant tortoises. However, the focus on age as a factor should be emphasized more throughout manuscript. Additionally, minor clarifications in sample numbers, methodology, and figure presentation would enhance clarity and coherence.

---

## Round 0.2 · Minor Revisions

Both reviewers agreed that the manuscript had improved from the previous version, but reviewer 2 in particular still had a couple of minor concerns, particularly around the age estimation of the tortoises.

Reviewer 1 ·

Basic reporting

Authors addressed all of my concerns. My main concern on using the GTDB database (which is based on whole genomes), is the possibility of a genome not including a ribosomal operon (due to incompleteness), meaning that the sequence will come back as unclassified mainly due to missing information in the database. It is OK to maybe just add a sentence to this effect in the results section if the authors would rather use this database. I appreciate running the analysis using SILVA. Was any of the "unclassified" taxa with the GTDB database now classified with SILVA?

Experimental design

All good here

Validity of the findings

All good here

Reviewer 2 ·

Basic reporting

Authors improved the manuscript sufficiently in this regard.

Experimental design

The age estimation explanation still needs to be improved. See more detailed response in the attachment.

Validity of the findings

Figures and tables are now much more informative.

Annotated reviews are not available for download in order to protect the identity of reviewers who chose to remain anonymous.

---

## Round 0.3 · accepted · Accept

The reviewer was happy that the changes you have made are sufficient to address their previous concerns, and so the manuscript is now ready for publication.

Reviewer 2 ·

Basic reporting

The authors have made solid and satisfying corrections in the manuscript and from my end it can be accepted to publish.

Experimental design

No comment.

Validity of the findings

No comment.